# Immune Activation and Inflammatory Response Mediated by the NOD/Toll-like Receptor Signaling Pathway—The Potential Mechanism of Bullfrog (*Lithobates catesbeiana*) Meningitis Caused by *Elizabethkingia miricola*

**DOI:** 10.3390/ijms241914554

**Published:** 2023-09-26

**Authors:** Fulong Li, Baipeng Chen, Ming Xu, Yang Feng, Yongqiang Deng, Xiaoli Huang, Yi Geng, Ping Ouyang, Defang Chen

**Affiliations:** 1Department of Aquaculture, College of Animal Science & Technology, Sichuan Agricultural University, Chengdu 611130, China; lifulong0318@163.com (F.L.); 18224417292@163.com (B.C.); xuming2021202076@163.com (M.X.); chengdf_sicau@126.com (D.C.); 2Department of Basic Veterinary, College of Veterinary Medicine, Sichuan Agricultural University, Chengdu 611130, China; fengyang_sicau@163.com (Y.F.); gengyisicau@126.com (Y.G.); ouyang.ping@live.cn (P.O.); 3Fisheries Institute, Sichuan Academy of Agricultural Sciences, Chengdu 611731, China; dyqhxl@126.com

**Keywords:** *Elizabethkingia miricola*, *Lithobates catesbeiana*, bacterial meningitis, NOD-like receptor signaling pathway, Toll-like receptor signaling pathway

## Abstract

*Elizabethkingia miricola* is an emerging opportunistic pathogen that is highly pathogenic in both immunocompromised humans and animals. Once the disease occurs, treatment can be very difficult. Therefore, a deep understanding of the pathological mechanism of *Elizabethkingia miricola* is the key to the prevention and control of the disease. In this study, we isolated the pathogenic bacteria from bullfrogs with dark skin color, weak limbs, wryneck, and cataracts. Via subsequent morphological observations and a 16S rRNA gene sequence analysis, the pathogen was identified as *Elizabethkingia miricola*. The histopathological and transmission electron microscopy analysis revealed that the brain was the main target organ. Therefore, brain samples from diseased and healthy bullfrogs were used for the RNA-Seq analysis. The comparative transcriptome analysis revealed that the diseased bullfrog brain was characterized by the immune activation and inflammatory response, which were mediated by the “NOD-like receptor signaling pathway” and the “Toll-like receptor signaling pathway”. We also performed qRT-PCR to examine the expression profile of inflammation-related genes, which further verified the reliability of our transcriptome data. Based on the above results, it was concluded that the NOD/Toll-like receptor-related networks that dominate the immune activation and inflammatory response were activated in the brain of *Elizabethkingia miricola*-infected bullfrogs. This study contributes to the search for therapeutic targets for bullfrog meningitis and provides basic information for establishing effective measures to prevent and control bullfrog meningitis.

## 1. Introduction

*Elizabethkingia miricola*, a non-fermenting rod-shaped bacteria, was first described in 2003 when it was isolated from condensation water on the space station Mir [1]. It was initially identified as *Chryseobacterium*, which was later reclassified and closely related to *Elizabethkingia meningoseptica* (previously *C. meningosepticum*) [2]. *Elizabethkingia miricola* has been demonstrated to be highly pathogenic to both humans and animals. In addition, whole-genome sequencing revealed that *Elizabethkingia miricola* isolated from animals had highly similarity with human clinical isolates [3]. It can cause meningitis [4], pneumonia [5], sepsis [6], and urinary tract infections [7] in newborns and immunodeficient people. Furthermore, *Elizabethkingia miricola* could penetrate the blood–brain barrier, causing meningitis and damage to the central nervous system, leading to high mortality and morbidity in frogs [8]. Therefore, it is necessary to investigate its pathogenic process in order to better prevent and control the occurrence of diseases caused by *Elizabethkingia miricola*.

Bacterial meningitis is an infectious disease with high morbidity and mortality worldwide, usually referring to the inflammation of the meninges after the bacteria penetrate the blood–brain barrier [9]. Studies have shown that increased blood–brain barrier permeability is a prerequisite for the bacterial penetration of the blood–brain barrier [10]. After bacterial invasion of the central nervous system, it can promote the production and release of various cytokines such as IL-1β, IL-8, IL-6, and TNF-α in the brain expressed by endothelial cells, astrocytes, microglia, and neurons [11], where these cytokines have been shown to be involved in the disruption of the blood–brain barrier, further increasing blood–brain barrier permeability and recruiting leukocytes to the site of the infection, which may be responsible for the inflammatory cascade response as well as irreversible neuronal injury [12]. However, during the development of bacterial meningitis, the release of cytokines, the initiation of the inflammatory cascade response, and the recruitment of immune cells are associated with multiple signaling pathways [13]. But, what signaling pathways mediate these processes in bullfrogs after the infection with *Elizabethkingia miricola*, which is a very common and key disease in its farm process, is not clear.

Bullfrog (*Lithobates catesbeianus*), a kind of aquaculture animal with important economic value, has been favored by many consumers since it was first introduced into China from Cuba in 1959 [14]. Bullfrog breeding in China was gradually promoted in the 1990s and has now spread throughout the country, where the annual production reached 600,000 tons of bullfrogs in 2021 [15]. However, due to the widespread distribution of *Elizabethkingia miricola* in the natural environment and the body of bullfrogs, the related disease broke out frequently, and the breeding of bullfrogs was heavily hampered, causing huge economic losses [16,17,18]. But, the mechanism caused by *Elizabethkingia miricola* is not fully understood, which makes the treatment of bullfrogs caused by *Elizabethkingia miricola* extremely difficult.

In this study, we aimed to isolate and characterize the causative pathogen from bullfrogs with suspected meningitis and used histology, molecular biology, and transcriptomics to classify and identify the pathogenic bacteria and explore its infection mechanism to the bullfrog brain, so as to understand the pathogenesis of *Elizabethkingia miricola* to bullfrogs deeply, which may provide targeted clues for treatment options for frogs.

## 2. Results

### 2.1. Isolation and Identification of Pathogenic Bacteria

Compared with the healthy bullfrog, the diseased bullfrog showed dark skin color and weak limbs (Figure 1A), wryneck (Figure 1B), and cataracts (Figure 1C). After necropsy, it was found that the diseased bullfrog had ascites in the abdominal cavity (Figure 1D). Moreover, the swelling of the liver with localized ischemia (Figure 1E). Furthermore, the intestine was free of food but filled with transparent fluid (Figure 1F).

The main strains isolated from diseased bullfrogs (brain, liver, spleen, and kidney) were consistent in their characteristics. The bacterium forming small colonies were round or oval, white, translucent, shiny, slightly elevated, with smooth edges and 1–2 mm in diameter (Figure 2A). The results of Gram staining under the microscope showed that the bacterial cells stained red, were arranged singly or in pairs, and appeared as short rods, indicating that the strain was a Gram-negative short rod (Figure 2B). A single band was amplified via polymerase chain reaction (PCR) using 16S rRNA gene primers. The 16S rRNA gene sequences of isolated bacteria were 99% similar to *Elibethkingia miricola* (ON714884.1). Moreover, a phylogenetic tree analysis showed that the bacteria isolated in this study clustered with *Elizabethkingia miricola* (Figure 2C).

### 2.2. Histopathological Changes

To observe the histopathological status of bullfrogs infected with *Elizabethkingia miricola*, H&E staining was performed using tissues from the brain, eye, kidney, heart, liver, intestine, spleen, and lung, respectively. According to the histopathological observations, the most obvious lesions were found in the brain and eye tissues of bullfrogs (Figure 3). The pathological changes such as massive cell proliferation in the meningeal layer of the affected frog, severe edema and thickening of the meninges with hemorrhage, severe edema of the brain stroma, widening of the pericellular gaps, showing extensive vacuolization, and the disappearance of some brain tissue cells (Figure 3B,C) are consistent with those of meningitis. Ocular tissue with inflammatory cell infiltration, severe scleral edema, and the hemolysis of some ocular vessels was also present (Figure 3E,F). In addition, the epithelial cells of the renal tubules were mildly enlarged and vacuolated, where some cells were necrotic, and the lumen of the renal capsule was dilated (Appendix A). Inflammatory cell infiltration was found in the heart, intestine, and spleen (Appendix A); the liver showed more brown pigmentation and the vacuolar degeneration of hepatocytes (Appendix A); and there was slight alveolar microvascular hemorrhage (Appendix A). Finally, based on the extent of lesions in the different organs of bullfrogs (organ index), the heatmap analysis showed that the bullfrog brain was the primary target organ for the *Elizabethkingia miricola* infection (Figure 3G).

To determine the ultrastructure of brain tissue after the *Elizabethkingia miricola* infection, electron microscopic observations were carried out, and the results showed that the neuronal cells were necrotic and vacuolated, and the intracytoplasmic mitochondria, rough endoplasmic reticulum, and ribosomes were indistinct or dissolved (Figure 4A); the blood vessels showed damage, the endothelial cells were vacuolated (Figure 4B), the mitochondria were swollen, and the cristae disappeared (Figure 4C); and the myelin sheath was structurally altered, and the axons within the myelin sheath were dissolved (Figure 4D).

### 2.3. Elizabethkingia Miricola Infection Affects Gene Expression Profile in the Brain of Bullfrog

To analyze the response of bullfrogs to the *Elizabethkingia miricola* infection at the genetic level, bullfrog brains were collected for the transcriptome analysis. In total, 173,754,624 and 189,805,604 raw reads were gained, respectively, from the control (Control-1, Control-2, and Control-3) and Eli (Eli-1, Eli-2, and Eli-3) samples using an Illumina Novaseq 6000 sequencing platform. After quality filtering and trimming, 54,541,358 to 65,771,358 reads were obtained for six samples, and the percentage of Q20 base was over 97.14% (Table 1). The obtained clear reads were mapped with the reference genome (Genome ref. number: GCA_002284835.2) using HISAT2 software (Version 2.1.0, http://ccb.jhu.edu/software/hisat2/index.shtml accessed on 18 March 2023), and the mapping rate was all higher than 78% (Appendix A).

Prior to the analysis of differentially expressed genes (DEGs), a principal component (PCA) and Venn analysis were performed. PCA plots showed significant differences between the control and Eli groups (Figure 5A). From the Venn plots, there are 9278 common genes expressed in both groups, 518 unique genes in the control group, and 840 unique genes in the Eli group (Figure 5B). For DEGs, the MA plots showed that 622 genes were upregulated and 602 genes were downregulated between the control and Eli groups (Appendix A and Figure 5C). Moreover, the hierarchical clustering heatmap allows for the gene expression patterns between the control and Eli groups (Figure 5D).

### 2.4. Immune Pathways Are Enriched in Brains after Infection with Elizabethkingia Miricola

The functions of DEGs were classified by using the Cluster of Orthologous Groups of proteins (COG), Gene Ontology (GO), and the Kyoto Encyclopedia of Genes and Genomes (KEGG) databases. The results showed that among the 21 COG classifications, the “function unknown” represents the largest group, and the “posttranslational modification, protein turnover, chaperones”, “intracellular trafficking, secretion, and vesicular transport”, and “Transcription” are next to it (Appendix A). Subsequently, the DEGs were annotated with GO, which showed that 3878 unigenes were categorized into 45 GO terms, of which “Cellular process”, “Cell part”, and “Binding” were dominant in the categories of “Biological process”, “Cellular component”, and “Molecular function”, respectively (Appendix A).

To characterize the biological pathways in the transcriptome, 309 unigenes were classified into 43 KEGG pathways which belonged to six categories, including genetic information processing, metabolism, cellular processes, human diseases, environmental information processing, and organismal systems. Among the 43 KEGG pathways, “Signal transduction” (172 unigenes) was attributed with the highest number of unigenes, and the next one was the “Immune system” (155 unigenes) (Figure 6A). The analysis of the first 20 enriched pathways showed the presence of six immune-related pathways, the “Complement and coagulation cascades”, “IL-17 signaling pathway”, “NOD-like receptor signaling pathway”, “Cytosolic DNA-sensing pathway”, “Toll-like receptor signaling pathway”, and “Neutrophil extracellular trap formation”, it is noted that the “NOD-like receptor signaling pathway” with the largest number of DEGs was being enriched (Figure 6B). In addition, we performed a chord plot analysis of the 10 most enriched GO terms, which showed that most of the GO terms with higher enrichment were related to immunity, such as the “defense response” and “inflammatory response” (Figure 6C). The clustering analysis of DEGs related to immunology showed significant differences in the expression patterns between the control and Eli groups (Figure 6D). Most immune-related genes were significantly upregulated in the Eli group compared to the control group. To explore the *Elizabethkingia miricola* on the regulative pathway of the brain inflammatory response, immunomodulatory key signaling molecule-related genes *NLRP*, *ASC*, *NAIP*, *NFκB*, *COX2*, *IL-8*, *TLR4*, and *CARD9* were investigated. The results showed that unique genes involved in the inflammatory responses are activated in the brain (Figure 6E). Taken together, this indicates that the *Elizabethkingia miricola* infection activated immunity and led to inflammation in the brains of bullfrog.

### 2.5. Activation of NOD/Toll-like Signaling Pathway after Infection with Elizabethkingia Miricola

The transcriptomic data was further analyzed. The KEGG enrichment analysis results indicated that the NOD/Toll-like signaling pathway were significantly enriched from the DEGs’ list (*p* < 0.05) (Figure 6B). More specifically, a large number of genes in the NOD/Toll-like signaling pathway were significantly upregulated, including the *TLR4* and *NFκB*, chemokines *IL-8* and *CXCL10*, and pro-inflammatory cytokine *IL-1β* (Figure 7A,B and Appendix A). The upregulated expression of these key genes may activate the signaling pathways in which they are located, resulting in an inflammatory response in the bullfrog brain after the *Elizabethkingia miricola* infection.

### 2.6. Validation of RNA-Seq Results via Quantitative PCR Analysis

To further validate the results from the RNA sequencing data, five genes related to the inflammatory response were selected upon performing qRT-PCR (Figure 8). The results showed that the expression trend of each gene was consistent with the high-throughput sequencing data. In addition, cytokines promoting inflammation were significantly upregulated in the Eli group, suggesting that inflammation was activated in the brain tissue infected with *Elizabethkingia miricola*.

## 3. Discussion

Phylogenetic and histological analyses showed that the pathogeny of the diseased bullfrog in this study was *Elizabethkingia miricola*, and the brain is one of the main target organs of bullfrogs infected with *Elizabethkingia miricola*. Subsequently, transcriptome studies were used to reveal the regulatory mechanisms of bullfrog meningitis caused by *Elizabethkingia miricola*, and the results suggested that the diseased bullfrog brain was characterized by immune activation and the inflammatory response mediated by the “NOD-like receptor signaling pathway” and the “Toll-like receptor signaling pathway”. We performed qRT-PCR to examine the expression profile of inflammation-related genes, which further verified the transcriptome results.

*Elizabethkingia miricola* is an emerging pathogen that is extremely difficult to treat because it shows an excellent resistance to antibiotics [19]. In this study, we found that the typical neurological symptoms of the bullfrogs include wryneck and cataracts. The histopathological analysis showed that the brain was the main target organ after the infection, and further ultrastructural observation revealed damage to the brain tissue vessels, severe neuronal necrosis, and mitochondrial swelling. Similar results have been demonstrated in other animals. *Elizabethkingia miricola* can cause symptoms such as wryneck and cataracts in *Quasipaa spinosa*, *Rana sylvatica*, and *Hyla viridis* [20,21]. In addition, transmission electron microscopy showed that *Elizabethkingia miricola* caused severe neuronal necrosis and mitochondrial swelling in the brain tissue of *Pelophylax nigromaculatus* [8]. The above results indicate that the brain is the main target organ of *Elizabethkingia miricola*, suggesting that *Elizabethkingia miricola* can cross the bullfrog blood–brain barrier, and thus, cause brain damage. However, the majority of antibiotics have difficulty crossing the blood–brain barrier, resulting in the brain not reaching effective therapeutic concentrations, which may explain why meningitis is difficult to treat. Therefore, further screening for antibiotics that readily cross the blood–brain barrier may be one way to control and treat frog meningitis.

The comparative transcriptome analysis has been widely used in different animals to provide reliable data on immune mechanisms. In the present study, bullfrogs infected with *Elizabethkingia miricola* showed significant changes in the brain transcriptome, which may be related to bacterial proliferation in the frog brain as well as the immunity of bullfrogs to bacterial attack. Based on the GO enrichment analysis, for those with high enrichment associated with immunity, such as the “defense response” and the “inflammatory response”, they are both biological responses in the immune system, and their activation may be a protective cellular response to the *Elizabethkingia miricola* infection. The KEGG enrichment analysis showed that immune correlation was also enriched, such as the “Complement and coagulation cascades”, “IL-17 signaling pathway”, “NOD-like receptor signaling pathway”, “Cytosolic DNA-sensing pathway”, “Toll-like receptor signaling pathway”, and “Neutrophil extracellular trap formation”. Most of them were also associated with an inflammatory response, which is consistent with the results of the GO enrichment analysis. In addition, these immune system pathways are also altered by pathogenic infections in other breeding animals. The enrichment of the “Complement and coagulation cascades”, “NOD-like receptor signaling pathway”, and “Toll-like receptor signaling pathway” is present in the brain transcriptome of *Elizabethkingia miricola*-infected black spotted frog [22]. The transcriptome analysis of the flounder showed that the immune system pathways, including the “Intestinal immune network for IgA production”, “Toll-like receptor signaling pathway”, “NOD-like receptor signaling pathway”, and “Cytosolic DNA-sensing pathway” were enriched in the spleen cells after the *Edwardiana* infection [23]. These studies suggest that the activation of the immune system may be a common outcome of different hosts following the bacterial infection.

Concerning pattern recognition receptors (PRRs), the recognition of pathogens is the first step in the host immune response. NOD-like receptors (NLRs) belong to a family of intracellular PRRs that recognize intracellular pathogens and initiate downstream signaling events [24,25,26]. NOD1 and NOD2 in the cytoplasm are key PRRs for anti-bacterial immunity, and they mediate the activation of the immune response signaling pathways by recognizing peptidoglycan, a component of the bacterial cell wall [27]. More specifically, signals via NOD1 and NOD2 induce inflammatory cytokines and other anti-microbial genes that contribute to host defense [28,29]. In addition, TLR4 is the major lipopolysaccharide (LPS) receptor among the toll-like receptors and TLR4 is activated by LPS. TLR4 can bind to MD-2 on the cell surface to form a complex that is required for the induction of inflammatory cytokines [30]. In our study, the “NOD-like receptor signaling pathway” and the “Toll-like receptor signaling pathway” served as the main pathways for KEGG enrichment, which may be the main way for the host recognition of *Elizabethkingia miricola* and the induction of the inflammatory response.

According to Kim’s description, the process of bacterial meningitis involves the mucosal colonization of pathogenic bacteria, the invasion of the blood, survival and replication in the blood, and the eventual crossing of the blood–brain barrier [31]. In this study, the ultrastructure of the brain of the diseased bullfrog was observed via transmission electron microscopy, which revealed the presence of damage to the blood vessels, indirectly proving that the pathway for *Elizabethkingia miricola* to enter the brain of the bullfrog was across the blood–brain barrier, which is consistent with the findings of Kim (Figure 9). Therefore, the pathway of *Elizabethkingia miricola* entry into the brain of bullfrogs can be hypothesized. However, the inflammatory response is one of the important factors causing the disruption of the blood–brain barrier permeability during the infection of meningitis-causing pathogens [32]. Moreover, the “NOD-like receptor signaling pathway” and the “Toll-like receptor signaling pathway” may be the pathway for the bullfrog host to recognize the bacteria and induce the inflammatory response. Therefore, we can hypothesize the gene regulatory network in bullfrog cells after the infection with *Elizabethkingia miricola* (Figure 10). In detail, *Elizabethkingia miricola* crossing the blood–brain barrier (BBB) into the brain induce the pattern recognition receptors (PRRs) to interact with pathogenic lipopolysaccharides (LPS) and peptidoglycans (PGN), inciting the further activation of inflammasomes and NFκBs via the “NOD-like receptor signaling pathway” and the “Toll-like receptor signaling pathway”, thus promoting the transcription and expression of IL-8, IL-1β, and CXCL10 and leading to bacterial meningitis, but further research is needed to confirm these speculations.

## 4. Materials and Methods

### 4.1. Bullfrogs Sample Collection

In this study, 20 diseased bullfrogs (Eli group) and 20 healthy bullfrogs (Control group) were obtained from farms in Chengdu (Sichuan Province, China) via multiple collections. The diseased bullfrogs showed dark skin color, weak limbs, wryneck, and cataracts. After collection, nine diseased bullfrogs and nine healthy bullfrogs were anesthetized with MS222 (Aladdin, Shanghai, China), and their tissues were collected. In a sterile environment, the brain, liver, spleen, and kidney of three diseased bullfrogs were inoculated in a brain–heart immersion medium and then cultured in an incubator at 28 °C. The remaining individuals were stained with hematoxylin and eosin (H&E), observed via transmission electron microscopy, and RNA was extracted as described below. Animal procedures were approved by the Animal Experiment Committee of Sichuan Agricultural University and carried out according to the relevant guidelines.

### 4.2. Isolation and Sequencing of Pathogenic Bacteria

After 48 h of bacterial incubation, individual colonies were selected and inoculated into a BHI liquid medium, placed in an oscillating incubator, and incubated for 24 h at 28 °C. Total genomic DNA was extracted from the isolates using bacterial DNA extraction kits (Foregene, Chengdu, China). The universal primers (27F: 5′-AGAGTTTGATCCTGGCTCAG-3′ and 1492R: 5′TACGGCTACCTTGTTACGACTT-3′) were used for the PCR amplification of the bacterial 16S rRNA gene. PCR products were then pooled and sent to a company (Tsingke Biotechnology, Beijing, China) for purification and sequencing (Sanger). Sequences were edited using the software Geneious (Version 2023.2 https://www.geneious.com/download/ (accessed on 18 March 2023)) to check the electropherograms. The Nucleotide BLAST analysis of the amplified sequences obtained in this study was performed on the website of the National Center for Biotechnology Information (http://www.ncbi.nlm.nih.gov/blast (accessed on 18 March 2023)). Clustal W software (https://www.genome.jp/tools-bin/clustalw (accessed on 18 March 2023)) was used for multi-sequence alignment, and MEGA 11.0 software was used to construct the phylogenetic tree via the neighbor-joining method with 1000 non-parametric bootstrap replicates.

### 4.3. Histological Analysis of Bullfrogs

Brain, eyes, kidney, heart, liver, intestines, spleen, and lung samples from healthy and diseased bullfrog groups (*n* = 3) were collected and fixed in 4% PFA for 24 h and dehydrated using a series of ethanol treatments, which was then cleared in xylene and embedded in paraffin wax. Sections were cut at a 4 µm thickness and stained with hematoxylin and eosin (H&E). Tissue slides were examined using an optical microscope (Nikon, Tokyo, Japan). The degrees of structural alterations, hemorrhages, deposits, edema, hyperplasia, infiltrations, and necrosis in the organs were evaluated according to the modified scoring system designed by Bernet et al. (Appendix A) [33]. Every change was assessed with a score (S) ranging from 0 to 6, depending on the degree and extent of the change: (0) unchanged, (2) mild change, (4) moderate change, and (6) severe change (diffuse lesion). Intermediate values were also considered. The organ index (Iorg=∑t∑alt[S×ωif]) of each experimental group were calculated (ωif: importance factor).

### 4.4. Transmission Electron Microscopy

The *Elizabethkingia miricola* infected brain was sampled for an ultrastructural examination. Briefly, the brain was fixed with 2.5% glutaraldehyde at 4 °C for 24 h, washed with PBS (pH 7.2), followed by postfixing in 1% osmium tetroxide, dehydrated with graded acetone, penetrated, and embedded, where an ultrathin section was obtained and subsequently stained with uranium acetate for 10 min and then with lead citrate for 1 min. Micrographs were taken with transmission electron microscopy (JEOL, Tokyo, Japan) operating at 80 kV.

### 4.5. RNA Isolation, cDNA Library Preparation, and Sequencing

Total mRNA was extracted from six brain tissues (three samples per group) using TRIzol^®^ Reagent according to the manufacturer’s instructions (Invitrogen, Carlsbad, CA, USA), and genomic DNA was removed using DNase I (TaKara, Kyoto, Japan). The RNA quality was determined using the 2100 Bioanalyser (Agilent, Santa Clara, CA, USA) and quantified using the ND-2000 (Thermo Scientific, Wilmington, DE, USA). Only a high-quality RNA sample (OD260/280 = 1.8~2.2, OD260/230 ≥ 2.0, RIN ≥ 6.5, 28S:18S ≥ 1.0, >1 μg) was used to construct the sequencing library. Poly (A) mRNA was isolated from the total RNA using poly (dT) oligo-attached magnetic beads, and the RNA-seq transcriptome library was prepared following the TruSeqTM RNA sample preparation kit from Illumina (San Diego, CA, USA) using 1 μg of total RNA. mRNA was randomly fractured via a fragmentation buffer and sifted out into small fragments of about 300 bp. Subsequently, under reverse transcriptase, random hexamers were added to synthesize the first-strand cDNA using mRNA as the template, followed by second-strand synthesis to form a stable double-strand structure. These double-stranded cDNA fragments were end-repaired by adding a single “A” base and a ligation of adapters. The six libraries were sequenced using an Illumina Novaseq 6000 platform, with 150 bp paired-end reads produced.

### 4.6. Transcriptome Quality Control and Gene Annotation

The raw paired-end reads were trimmed, and the quality was controlled via SeqPrep (https://github.com/jstjohn/SeqPrep, accessed on 18 March 2023) and Sickle (https://github.com/najoshi/sickle, accessed on 18 March 2023) with default parameters. Then, the clean reads were separately aligned to the reference genome with the orientation mode using HISAT2 software (Version 2.1.0, http://ccb.jhu.edu/software/hisat2/index.shtml, accessed on 18 March 2023) [34]. The mapped reads of each sample were assembled via StringTie (Version 2.1.2, https://ccb.jhu.edu/software/stringtie/index.shtml?t=example, accessed on 18 March 2023) in a reference-based approach [35]. The unigenes were compared with known transcripts from six databases to obtain the annotated information, including Gene Ontology (GO) [36], the Kyoto Encyclopedia of Genes and Genomes (KEGG) [37], the Clusters of Orthologous Groups of proteins (COG), the National Center for Biotechnology Information (NCBI) non-redundant protein sequences (NR), Swiss-Prot, and The Pfam protein families (Pfam) [38].

### 4.7. Differential Gene Expression and Enrichment Analysis

To identify DEGs between the control and Eli groups (three replicates in each group), TPM (transcripts per million) were used to assess the gene expression. RSEM (Version 1.3.3, http://deweylab.biostat.wisc.edu/rsem/, accessed on 18 March 2023) was used to quantify gene abundances [36]. Essentially, the differential expression analysis was performed using the DESeq2 where Q value ≤ 0.05, DEGs with |log2FC| > 1 and Q value ≤ 0.05 were considered to be significantly differentially expressed genes [39]. In addition, the functional-enrichment analysis including GO and KEGG were performed to identify which DEGs were significantly enriched in GO terms and the metabolic pathways at the Bonferroni-corrected *p*-value ≤ 0.05 compared with the whole-transcriptome background. GO functional enrichment and the KEGG pathway analysis were carried out via Goatools (Version 0.6.5, https://github.com/tanghaibao/Goatools, accessed on 18 March 2023) and KOBAS (Version 2.1.1, http://kobas.cbi.pku.edu.cn/home.do, accessed on 18 March 2023) [40].

### 4.8. qRT-PCR Validation of DEG Expression

To verify the reliability of the DEG data, five inflammatory response-related genes were randomly chosen for the qRT-PCR analysis. Briefly, qRT-PCR was performed in a total volume of 10 μL containing 5 μL of TB Green™ Premix Ex Taq™ II, 0.2 μL of Rox, 1 μL of cDNA, 0.8 μL of each primer (Appendix A), and 3 μL of double distilled water. The reaction conditions used were as follows: 95 °C for 3 min, followed by 39 cycles of 95 °C for 10 s, 57 °C for 20 s, and 72 °C for 20 s, with the dissolution curve increasing from 0.5 °C to 95 °C every 5 s. Specific primers were designed on the basis of NCBI Primer-BLAST (NCBI, Bethesda, MD, USA), and the sequences are provided in Appendix A. All qRT-PCR reactions were performed in triplicate, and target specificity was determined based on the dissociation curve analysis. β-actin was selected as the internal control to normalize the expression level of each gene. The relative expression level of the target gene versus the β-actin gene was calculated using the 2^−ΔΔCT^ method.

### 4.9. Statistical Analyses

In this study, statistical difference was analyzed via SPSS 27.0 software (IBM Corp., Chicago, IL, USA). Charts were drawn via GraphPad Prism (Version 9.5.0) and Adobe Illustrator software (Version 26.0). A significant difference was determined using the one-way ANOVA analysis, and the significant level was set as *p* < 0.05 (*), *p* < 0.01 (**) or *p* < 0.001 (***).

## 5. Conclusions

In conclusion, this investigation demonstrated that, firstly, the pathogenic bacteria isolated from the brains of bullfrogs suspected of having meningitis were identified as *Elizabethkingia miricola*. Secondly, the potential mechanism of the *Elizabethkingia miricola* infection of bullfrog brains may be the immune activation and inflammatory response mediated by the NOD/Toll-like receptor signaling pathway.

## Figures and Tables

**Figure 1 ijms-24-14554-f001:**
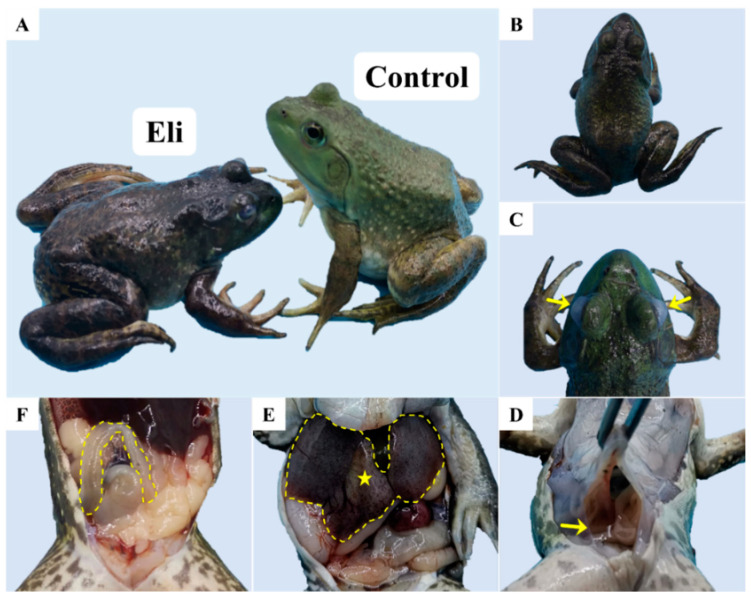
Clinical signs of diseased bullfrogs. (**A**) Left: dark body color and weak limbs (Eli), right: healthy frog (Control); (**B**) head tilted to one side; (**C**) white eye cornea and cloudy eye with mild protrusion (yellow arrow); (**D**) fluid accumulation (ascites) in the abdominal cavity (yellow arrow); (**E**) slight ischemia (yellow pentagram) of the liver (yellow dotted line); and (**F**) no food in the intestine, filled with transparent fluid (yellow dotted line).

**Figure 2 ijms-24-14554-f002:**
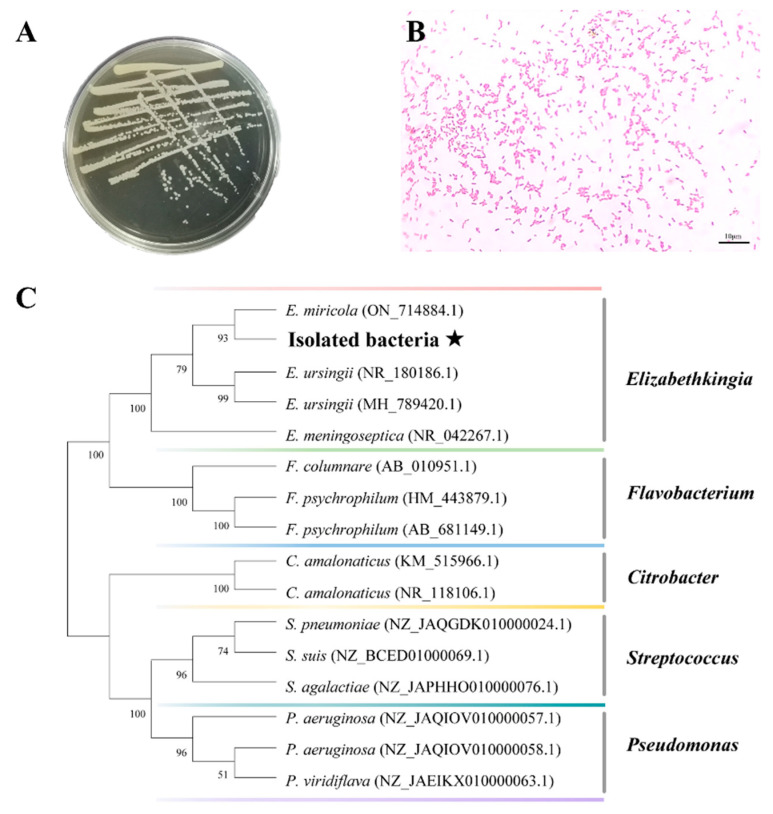
Isolation and characterization of pathogenic bacteria. (**A**) Growth characteristics of the isolates in BHI medium; (**B**) Gram staining of the dominant colonies at 1000×; and (**C**) phylogenetic tree of 16S rRNA sequences of the isolates and related strains constructed based on the neighbor-joining method. The number at each node indicates the percentage of bootstrapping in 1000 replicates. The black pentagram represents the sequence of pathogenic bacteria isolated in this study.

**Figure 3 ijms-24-14554-f003:**
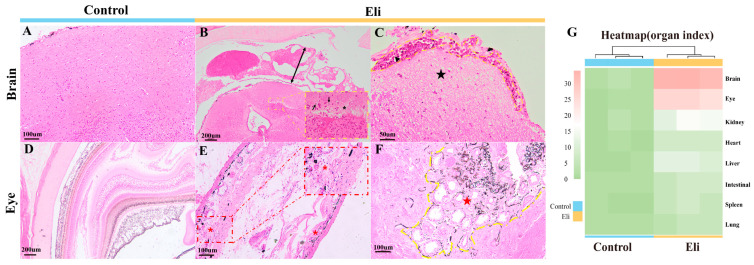
Histopathological changes in bullfrogs infected with *Elizabethkingia miricola*. (**A**) Control bullfrog brain. (**B**,**C**) Massive cell proliferation in the meningeal layer, severe edematous thickening of the meninges (double arrow) with hemorrhage (non-tailed arrow), loss of brain tissue cells (asterisk), enlargement of the intercellular spaces (tailed arrow), and widespread cellular cytoplasmic swelling (pentagram). (**D**) Control bullfrog eyes. (**E**,**F**) Infiltration of inflammatory cells in the eye tissue (asterisk). The sclera is severely edematous (yellow dotted line) and some of the ocular vessels appear hemolyzed (pentagram). (**G**) Overall health status (organ index) of frogs in different groups, based on histopathological lesions.

**Figure 4 ijms-24-14554-f004:**
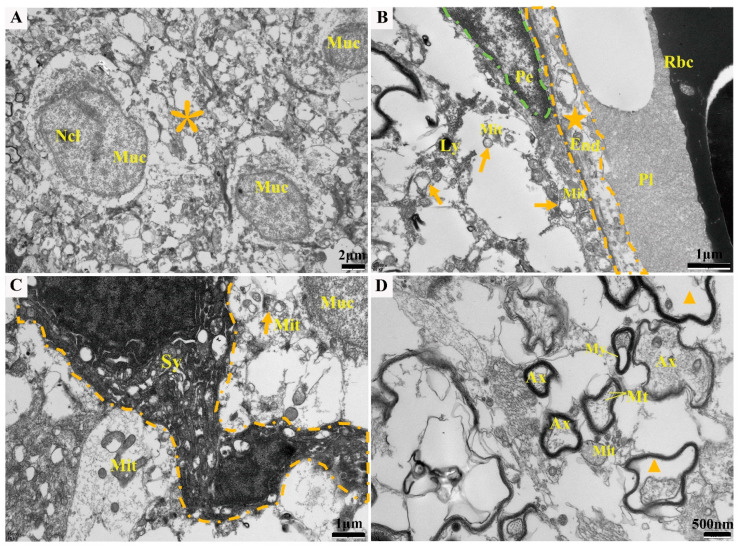
Ultramicropathological changes in brain tissue of affected frogs (**A**–**D**), including neuronal cell necrosis and cytoplasmic vacuolization (asterisks), vascular damage (pentagram), mitochondrial swelling (arrow), axonal lysis in the myelin sheath (triangles). Ncl: nucleolus. Nuc: nucleus. Rbc: red blood cell. Pi: plasma. End: endothelial cell. Pe: pericyte. Mit: mitochondrion. Ly: lysosome. Sy: synapse. My: myelin. Ax: axon. Mt: microtubule.

**Figure 5 ijms-24-14554-f005:**
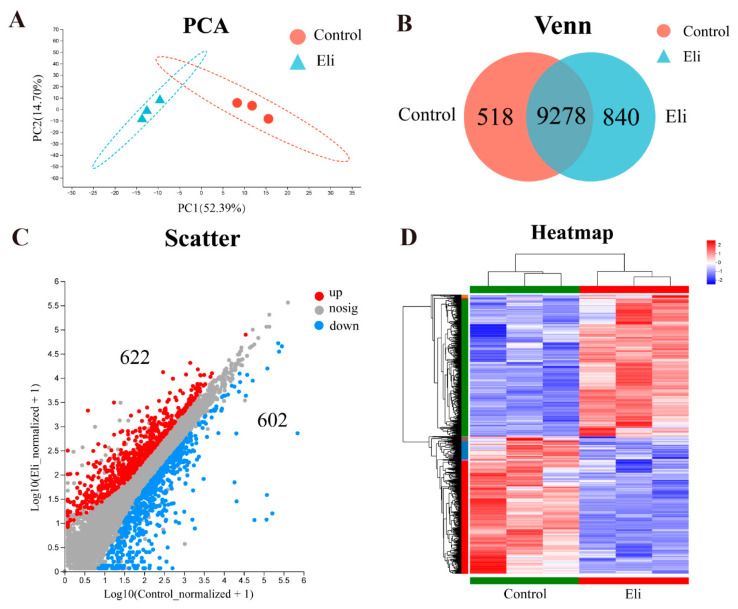
Analysis of differential expression genes (DEGs). (**A**) Principal component analysis (PCA) of the brain transcriptome in control and Eli groups. (**B**) Venn plots of DEGs among control and Eli groups. (**C**) MA plots showing DEGs in both groups. (**D**) Hierarchical clustering heatmap.

**Figure 6 ijms-24-14554-f006:**
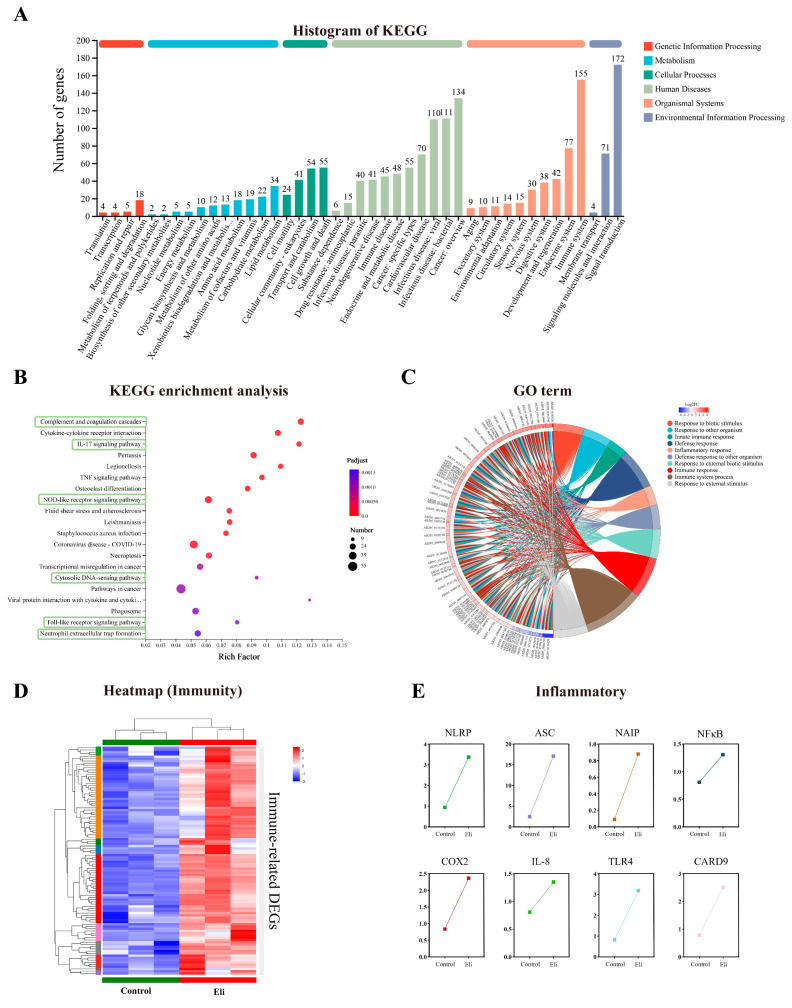
GO and KEGG annotation and enrichment analysis of differential expression genes (DEGs). (**A**) KEGG annotation of DEGs. (**B**) KEGG enrichment analysis was performed for differentially expressed genes. (**C**) Chord plot indicating the relationship of the most involved genes and the top 10 terms of GO. (**D**) Heatmap of immune-related DEGs. (**E**) Expression patterns of DEGs associated with inflammatory cytokine in the KEGG enriched pathway.

**Figure 7 ijms-24-14554-f007:**
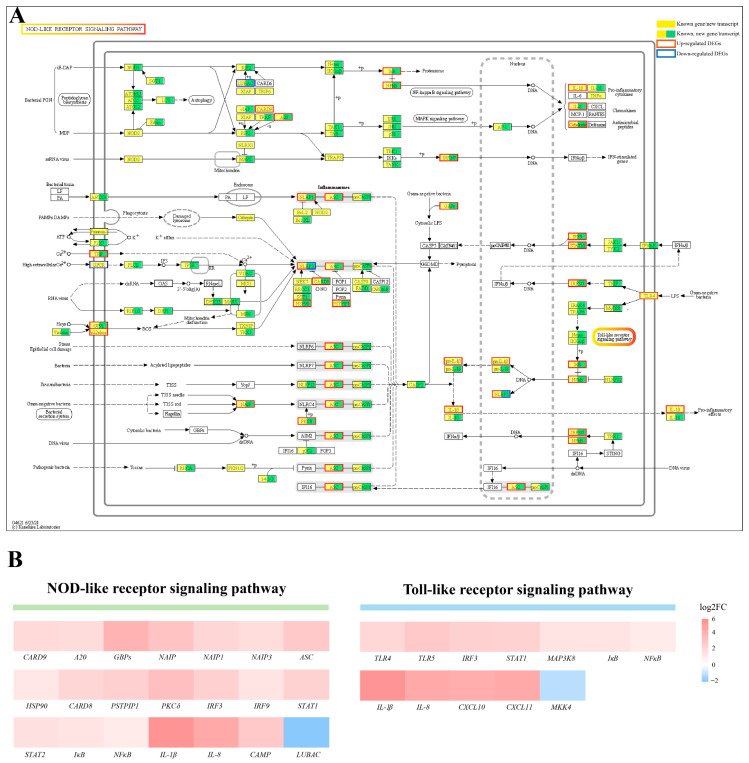
DEGs in NOD/Toll-like receptor signaling pathway. (**A**) NOD-like receptor signaling pathway. (**B**) Quantitative analysis of DEGs in NOD/Toll-like receptor signaling pathway. The data are from transcriptome, and the value is expressed in Log2FC (Eli/Control).

**Figure 8 ijms-24-14554-f008:**
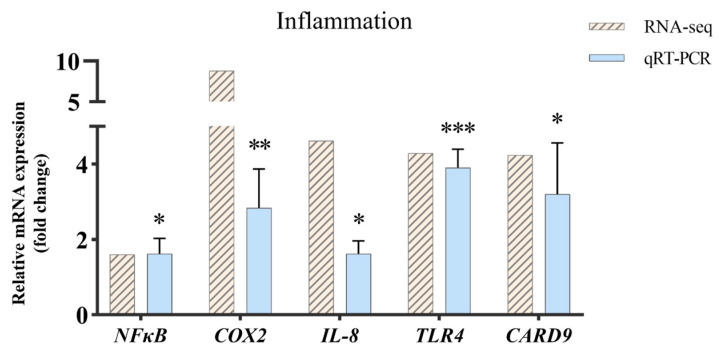
Validation of RNA-seq data by using real-time qRT-PCR. One-way ANOVA plus Bonferroni post-tests, *p* < 0.05 (*), *p* < 0.01 (**), *p* < 0.001 (***).

**Figure 9 ijms-24-14554-f009:**
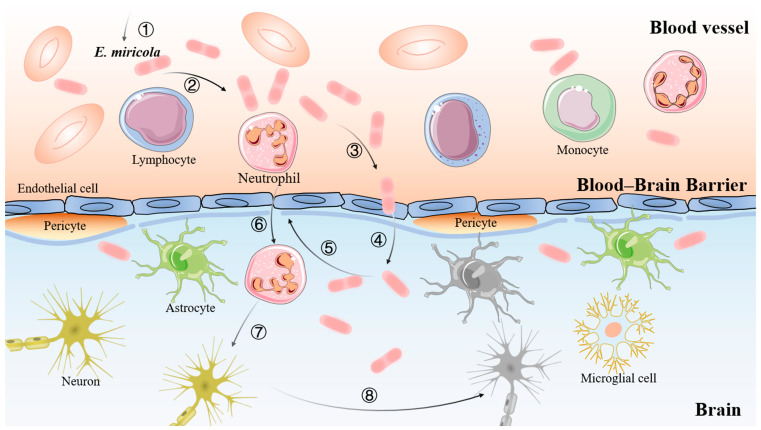
Schematic representation of the interaction of *Elizabethkingia miricola* with the brain of a bullfrog. (1): Entry of *Elizabethkingia miricola* into the blood circulation; (2): Proliferation of *Elizabethkingia miricola* in the bloodstream; (3): Crossing of the blood–brain barrier by *Elizabethkingia miricola*; (4): Invasion of *Elizabethkingia miricola* into the brain and the central nervous system; (5): Increase in the permeability of the blood–brain barrier; (6): Entry of inflammatory cells into the brain tissue; (7): Release of proinflammatory factors by the leukocytes and inflammatory cells that have entered the CNS and acted on the neurons; and (8): Neuronal damage.

**Figure 10 ijms-24-14554-f010:**
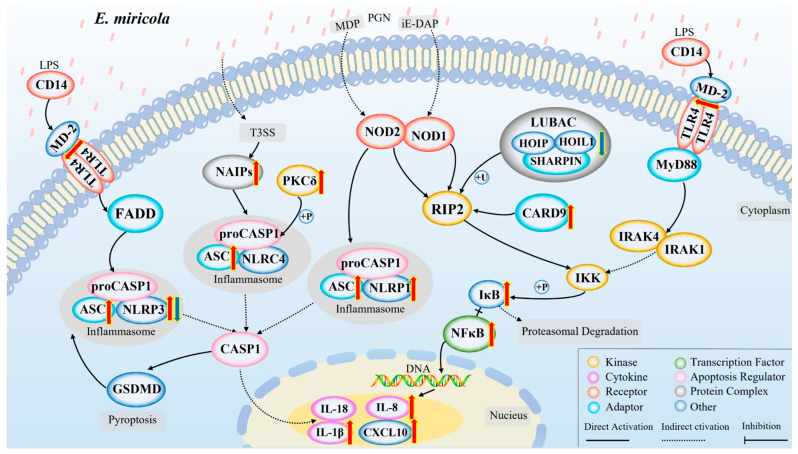
A model of inflammatory response in brains of bullfrogs infected with *Elizabethkingia miricola*. A schematic diagram showing the main inflammatory response events involved in brain *Elizabethkingia miricola* in bullfrogs inferred from the results of this study. Red arrow represents upregulation; blue arrow represents downregulation.

**Table 1 ijms-24-14554-t001:** Quality control information for transcriptome samples.

Samples	Raw Reads	Raw Bases	Clean Reads	Error Rate (%)	Q20 (%)
Control-1	55,240,800	8,341,360,800	54,645,342	0.0259	97.71
Control-2	55,210,162	8,336,734,462	54,541,358	0.0265	97.51
Control-3	63,303,662	9,558,852,962	62,634,710	0.0262	97.64
Eli-1	66,743,592	10,078,282,392	65,771,358	0.0274	97.14
Eli-2	60,743,954	9,172,337,054	59,997,100	0.0268	97.36
Eli-3	62,318,058	9,410,026,758	61,569,246	0.027	97.29

## Data Availability

The data presented in this study are deposited in the Sequence Read Archive (SRA) at the NCBI repository, accession number: PRJNA976064.

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
