# Peer review of "Immune Activation and Inflammatory Response Mediated by the NOD/Toll-like Receptor Signaling Pathway—The Potential Mechanism of Bullfrog (Lithobates catesbeiana) Meningitis Caused by Elizabethkingia miricola"

_ijms, 2023, doi:10.3390/ijms241914554_

Round 1

Reviewer 1 Report

Abstract: The abstract is written clearly. The usage of the word pathogen is redundant in the opening statement and can be changed.

Line 28: Change the word dominant to dominate.

The manuscript needs to be proof read by authors prior to submission. 

Please check for grammatical errors, sentence construction and spell errors.

This manuscript needs moderate amount of English language edits.

I request for the manuscript to be meticulously edited, corrected and re submitted. 

Author Response

Thank you very much for your comments and suggestions! Please see the attachment.

Reviewer 2 Report

This manuscript submitted by Li et al explores the pathological changes and signaling pathways affected by Elizabethkingia miricola infection in bullfrogs. The authors successfully isolated the pathogenic bacteria from bullfrogs and confirmed its identity as Elizabethkingia miricola through gene sequencing and phylogenetic analysis. They conducted histopathological analysis, which revealed that the brain is the primary target of this pathogenic bacteria. Additionally, the authors performed RNA sequencing analysis to investigate changes in gene expression between healthy and diseased frogs. By employing GO and KEGG analysis, they identified immune activation and inflammatory responses mediated by NOD/Toll-like receptor as key targets of Elizabethkingia miricola infection. Overall, this manuscript provides valuable insights into the altered gene expression associated with infection by Elizabethkingia miricola. The experiments were well-planned and executed. However, the manuscript is poorly organized with no proper references of figures, citations, and duplicate figures.

Major Comments:

1.      The study lacks significance as it only identifies differential gene expressions. The connection between these genes and disease progression is missing. It is unclear whether these genes act directly or indirectly.

2.      None of the figures are referenced in the main text, making it difficult to understand their context and interpretation.

3.      Figure 5 appears to be duplicated multiple times.

4.      The results section lacks references and displays "Error! Reference source not found." It is challenging to determine which references correspond to specific texts.

5.      The conclusion is not adequately supported by the results and data presented.

6.      The abstract mentions the research providing a reference for the prevention and control of Elizabethkingia miricola and the treatment of meningitis. However, these aspects are not discussed in the manuscript.

7.      Figures 7A, 9, and 10 appear to be adapted from external sources. Proper citations are required for these figures.

Editing of English language is required

Author Response

(The authors gave the same response as above.)

Reviewer 3 Report

This is an original study with some valid results and conclusions. Some areas require improvement

1) Figure 5 on pages 7-9 is confusing

2) Figure 5C is not a volcano plot (log fold change vs P -value) , this should be clarified

3) The main results should incorporate a table with the top upregulated and downregulated transcripts, their log FC and p-value

4) is y-axis on fig 8 showing log2 fc for rna seq data?

Author Response

(The authors gave the same response as above.)

Reviewer 4 Report

In the present manuscript, "Immune activation and inflammatory response mediated by 1 NOD/Toll-like receptor signaling pathway-The potential mechanism of the bullfrog (Lithobates catesbeiana) meningitis caused 3 by Elizabethkingia miricola" Fulong Li et. al. have shown that NOD/Toll-like receptor signaling pathway have been activated in the diseased bullfrogs. Here are some points I need to be considered:

 1.  In line 19 “we isolated the causative agent from bullfrogs with darkness, weakness of limbs, wryneck and cataracts”. Is the bullfrog showing these symptoms? If so, please rewrite the sentence.

2. Figure 6D, it’s not readable. Please explain what the author means by Immunity.

3. In Figure 8, please mention the statistical analysis that has been used.

4. For all the figures generated using software should be cited in the Figure legend.

5. Line 350, The diseased bullfrogs mainly showed darkness……., change it to dark skin color.

6. Line 359, After 48 h of bacterial incubation……, it’s incubation or infection?

Moderate editing is required throughout the manuscript. 

Author Response

(The authors gave the same response as above.)

Reviewer 5 Report

In this article by Li, Chen, and Xu et al., they isolated bacteria from diseases bullfrogs showing symptoms such as dark skin, cataracts, weak limbs, and wryneck. They found the causative agent from brain samples to be Elizabethkingia miricola via 16s rRNA sequencing. Using transcriptomics, they identified that inflammatory genes were upregulated in the diseased frogs. The authors also predicted some pathways which may be causing the symptoms and suggested that those pathways can be looked in detail to obtain a target for meningitis in frogs. The article was well-organized, the introduction was scope-specific, and methods well explained. Here are some minor comments:

Line 30-32: “provided a reference” does not seem to be the perfect phase to signify the importance of this research in this sentence. Maybe a more specific significance can be provided as a concluding statement of the abstract.

Line 37: “rod- shaped bacteria” would be a better phrase here.

Line 87: was there any difference in the morphology or content of proximal vs distal intestine?

Line 105: Can a higher magnification and better resolution image be used to replace this (1000X)? Sometimes a string of sphere-shaped bacterium looks similar to the image.

Line 128: In figure 3, I would suggest adding histopathological images of control bullfrog to show a relative comparison visually. Also, in (E) kindly describe in figure caption what the 0-30 scale represent.

Line 159: Line Table 1, make sure to change the caption.

Line 161: Expand DEGs.

Line 162: Expand PCA.

Figure 5: There is barely 1 sentence describing each part of Figure 5 in the manuscript text. It will be good to have more elaborate description of the data in the results section. Also, seems like there is some formatting error which makes the same figure appear multiple times.

Line 185: Expand COG.

Line 188: Expand GO.

Figure 6D: This part of the image is not clear enough to understand. This may be added as a separate figure or as a supplemental figure in a larger, high-resolution form.

Line 212: Mention the names of the genes here.

Line 230 and 246: Italicize Elizabethkingia miricola.

Line 246-258: This portion is a hypothesis and does not have in vitro/in vivo data supporting it. Hence, I believe it should not be included in the results section but may be included in the discussion section. Also, in line 253-258, it is important to reframe the sentences in a way to suggest that these are currently hypotheses and need more supporting data to confirm.

Figure 9: This figure does not contain any new information and hence can be included in the introduction.

Figure 10: Italicize bacteria name in the figure.

Line 279: Kindly try to avoid using strong verbs like “shows” and use verbs like “suggests” instead for results which have not been confirmed via other in vitro or in vivo techniques.

Line 305-319: Instead of restating the GO terms, it is essential to discuss their co-relation and how the bacterial may be activating the signaling pathway finally leading to the predicted response. Then some literature references with similar findings or which describes transcriptomic study with similar bacteria/host can be discussed. Overall, the discussion section can end with limitations of this study, future studies to bolster the suggestions provided in this article, and a big picture impact/significance of investing in this article/topic.

Line 347: This method may be divided into 2 portions- bullfrog sample collection; Isolation and sequencing of pathogenic bacteria.

Line 351: What was the criteria of this selection (9 out of 20)?

Line 388-390: mention incubation time and concentrations.

The quality of English language can be modified by checking grammatical errors through a software and rephrasing certain unclear sentences throughout the article. However, extensive language modification is only needed in the "Discussion" section of the article.

Author Response

(The authors gave the same response as above.)

Round 2

Reviewer 2 Report

The authors addressed all the raised concerns

Author Response

Thank you very much for all your comments and suggestions!